# Urea Formaldehyde Resin Resultant Plywood with Rapid Formaldehyde Release Modified by Tunnel-Structured Sepiolite

**DOI:** 10.3390/polym11081286

**Published:** 2019-08-01

**Authors:** Xiaona Li, Qiang Gao, Changlei Xia, Jianzhang Li, Xiaoyan Zhou

**Affiliations:** 1College of Materials Science and Engineering, Nanjing Forestry University, Nanjing 210037, China; 2Jiangsu Key Open Laboratory of Wood Processing and Wood-Based Panel Technology, Nanjing 210037, China; 3MOE Key Laboratory of Wooden Material Science and Application, Beijing Forestry University, Beijing 100083, China

**Keywords:** wood adhesive, tunnel-structured, sepiolite, rapid formaldehyde release

## Abstract

In order to reduce the cost of plywood and save edible resources (wheat flour), a cheap and resourceful clay, sepiolite, was used to modify urea formaldehyde (UF) resin. The performances of filler-filled UF resins were characterized by measuring the thermal behavior, cross section, and functional groups. Results showed that cured UF resin with SEP (sepiolite) formed a toughened fracture surface, and the wet shear strength of the resultant plywood was maximum improved by 31.4%. The tunnel structure of SEP was beneficial to the releasing of formaldehyde, as a result, the formaldehyde emission of the plywood bonded by UF resin with SEP declined by 43.7% compared to that without SEP. This study provided a new idea to reduce the formaldehyde emission, i.e., accelerating formaldehyde release before the product is put into use.

## 1. Introduction

Urea formaldehyde (UF) resin and its modified products have been widely used in the plywood industry because of many advantages, such as colorlessness, fast curing, water solubility, and low cost [1,2,3]. However, UF resin and its bonded panels have a serious formaldehyde emission issue [4]. Researchers have focused on chemical modification and optimizing the synthetic process to reduce the formaldehyde emission, by means such as reducing the formaldehyde/urea (F/U) molar ratio [5], using melamine to replace part of the urea [6], or adding scavengers [7,8]. These methods successfully reduced the free formaldehyde content of UF resin to a low level and yielded environmentally friendly panels. In recent years, protein-based adhesives have been developed to replace the formaldehyde-based adhesives [9,10], however, their industrial application was limited by some disadvantages, such as low solid content, low dry bonding strength, and high cost [11,12]. Therefore, UF resin will still be the predominant adhesive in the manufacture of plywood for some time [13,14].

During the fabrication of plywood, 20–30% of wheat flour (WF) is introduced into UF resin to increase its viscosity for good workability and pre-press property and to prevent the UF resin from over penetration into wood [15]. About 2 million tons of WF are used in plywood production every year, which is an appalling waste of precious food resources. In order to reduce the cost of plywood and save edible resources, many types of materials have been studied as substitutes for WF, e.g., chestnut shell powder [8] or hydrolyzed corn cob powder [16]. However, compared with flour, the initial viscosity of other types of filler-based adhesives needs to be further improved. In recent years, inorganic clay minerals present a considerable advantage in the wood adhesive modification area owing to their low price and large output. Most studies have thus far focused on two-dimensional disk-like clays such as montmorillonite [17,18,19], identifying their exfoliation mechanism and properties. However, one-dimensional rod-like fillers (e.g. sepiolite [20] and attapulgite [21]) have recently gained attention because these fillers occupy less surface space than the disk-like ones, allowing fillers to be evenly distributed in polymeric matrices [22].

Sepiolite (SEP) is a clay mineral with the formula Mg_4_Si_6_O_15_(OH)_2_·6H_2_O. This clay mineral has a nanometer tunnels structure showing a micro-fibrous morphology with a particle size in the 2–10 μm length range [23]. The specific surface area of SEP is close to 320 m^2^/g that is expressed in its admirable absorptive properties [24]. SEP is used to reinforce polymers because of the large interface area, which will result in a strong interaction between the polymer matrix and the nanofiller [25]. The studies using SEP as a filler of UF resin are relatively scarce. The unique tunnel structure of SEP can not only increase the adsorption area but also provide a channel for gas escape, and, therefore, we assumed that the unique tunnel structure of SEP will play a role in reducing formaldehyde emissions of UF resin resultant plywood.

In this study, different proportions of SEP were used to partially substitute WF as a UF resin filler in the plywood preparation process. Three-ply plywood was fabricated, and its wet shear strength and formaldehyde emission were measured. The effect of SEP addition on resin’s functional groups, cross-sections of cured resins, and thermal behavior of resins were evaluated by Fourier-transform infrared spectroscopy (FTIR), scanning electron microscope (SEM), and thermogravimetric analysis (TGA), respectively.

## 2. Materials and Methods

### 2.1. Materials

Urea (industrial grade, 98%) was purchased from the Henan Zhongyuan Chemical Company, China. Formaldehyde (aqueous solution, industrial grade, 37%) was purchased from the Guangdong Xilong Chemical Factory, China. Sodium hydroxide (solid, analytical grade, 95%), formic acid (aqueous solution, analytical grade, 98%), and ammonium chloride (powder, analytical grade, 99%) were purchased from the Beijing Chemical Factory, China. WF was obtained from the Beijing Guchuan Flour Company, China. Poplar (*Populus tomentosa* Carr.) veneers having 8% moisture content were purchased from Hebei province, Langfang, China. SEP (moisture content ≤2%, 1 g/cm^3^, 200–300 mesh) was purchased from the QiLiPing sepiolite factory, Nanyang, China.

### 2.2. Preparation of UF Resin

The UF resin was synthesized by formaldehyde and urea at an F/U molar ratio of 1:1 in the laboratory following the traditional “alkali–acid–alkali” three-step procedure as described previously in our research group [26,27]. The urea was added into formaldehyde solution three times at a mass ratio of 2.97:1.75:1. The formalin was placed in the reactor, adjusted to pH 8.0 with aqueous NaOH (30 wt %), and then the first amount of urea was added. The mixture was then heated to 90 °C for 1 h. The acidic reaction was brought by adding formic acid (20 wt %) to obtain a pH of 5.0 or so, and the condensation reactions were carried out until it reached a target viscosity. Then the mixture was adjusted to pH 8.0 by using NaOH, and the second amount of urea was added. After 30 min at 80 °C, the third amount of urea was added for a further stirring at 70 °C for 30 min. Then, the UF resin was cooled to room temperature, followed by adjusting the pH to 8.0.

Subsequently, 100.0 g of UF resin was added into a plastic cup followed by 0.6 g ammonium chloride; then SEP-based fillers (25 g) were added and stirred for 10 min for uniform mixing. The formulations of the fillers are shown in Table 1.

### 2.3. Preparation of Three-Ply Plywood

Poplar veneers with dimensions of 400 mm × 400 mm × 1.5 mm were used to prepare three-ply plywood. The resin was applied on double sides of a veneer at a spread rate of 300 g/m^2^. The resin-coated veneer was then stacked between two uncoated veneers with the grain directions of two adjacent veneers perpendicular to each other. The stacked plywood was hot-pressed at 1.0 MPa and 120 °C for 6 min and then preserved at room temperature for 24 h, before evaluation of wet shear strength and formaldehyde emission.

### 2.4. Wet Shear Strength Measurement

The wet shear strength of plywood was measured in accordance with the National Standard of GB/T 17657-2013 (GB/T 17657, 2013). For each adhesive, eight replicates (25 mm × 100 mm) were soaked in 63 ± 2 °C water for 3 h and then cooled to room temperature for 10 min before measurement. The average value of the wet shear strength was calculated and recorded.

### 2.5. Formaldehyde Emission Measurement

The formaldehyde emission of plywood was determined using the desiccator method in accordance with the procedure described in China National Standard GB/T 17657-2013. The plywood was cut into dimensions of 50 mm × 150 mm. Ten specimens of each adhesive were directly put into a 9–11 L sealed desiccator at 20 ± 2 °C for 24 h. The emitted formaldehyde was absorbed by 300 mL deionized water in a container. The formaldehyde concentration in the sample solution was determined using acetyl acetone–ammonium acetate solution and the acetyl acetone method, with colorimetric detection at 412 nm. For purposes of comparison, ten specimens of each adhesive were placed indoor for 30 days and then the formaldehyde emission was measured according to the same method described above. Finally, the formaldehyde emission decline rate of the plywood was calculated using Equation (1):*Decline rate = (initial − treatment)/original*(1)
where initial is the formaldehyde emission of the specimens tested directly, and treatment is the formaldehyde emission of the specimens after 30 days.

### 2.6. Fourier-Transform Infrared (FTIR) Spectroscopy

The FTIR spectroscopy (Nicolet FTIR spectrometer series 6700), in transmittance mode, was used for the characterization of the functional groups of the resins. The spectra were obtained in the spectral area 400–4000 cm^−1^ with a resolution of 4 cm^−1^ and 32 scans using potassium bromide (KBr) disc. Discs were prepared by mixing 1 mg of cured resins powder with 70 mg of KBr. Cured resins were prepared by drying the liquid resins in a convection oven at 120 °C for 2 h and then grinding them into a powder.

### 2.7. Thermal Stability of Cured Resins

Samples were placed in an oven at 120 °C until reaching a constant weight and then ground into powder. The stability of the cured adhesive was tested using a TGA instrument (TA Q50, WATERS Company, New Castle, DE, USA). About 5 mg of the ground powder was placed in a platinum tray and scanned from room temperature to 600 °C at a heating rate of 10 °C/min.

### 2.8. Scanning Electron Microscopy (SEM)

Samples were placed into a piece of aluminum foil and dried in an oven at 120 ± 2 °C until a constant weight was achieved. The sample fracture surface was sputter-coated with gold by using an ion sputter (Hitachi E-1010 Ion Sputter, Japan), and then a Hitachi S-3400N (Hitachi Science System, Ibaraki, Tokyo, Japan) scanning electron microscope was used to observe the fracture surface of the resultant adhesives.

### 2.9. Water Contact Angle Tests

The liquid Resin A5 and A1 in the same amount was uniformly coated on glass slides and then placed in an oven at 120 ± 2 °C for 2 h to form a cured resin film. The surface contact angles of the resin films were measured with a water contact angle (WCA) goniometer (DataPhysics Co. Ltd. Filderstadt, Germany) (OCA-20, DataPhysics Co. Ltd. Filderstadt, Germany). The measurements were performed via the sessile droplet method under standard parameters such as room temperature and drop volume (3 μL). At least three measurements of the angles on both film sides were recorded at an interval of 0.1 s for 180 s.

## 3. Results and Discussion

### 3.1. Wet Shear Strength Measurement

The wet shear strength of the plywood bonded by UF resin with different fillers is shown in Figure 1. All the values of wet shear strength of the plywood were above 0.7 MPa (labeled as the dash line) except the one of Resin A6. WF could absorb water and swell in the UF resin, which could prevent the UF resin molecule penetrating into the wood surface. Compared with WF, SEP does not swell in the resin, so that, using 100% SEP filler led to the resin’s excessive penetration into the veneer. Therefore, the wet shear strength of plywood bonded by Resin A6 decreased by 44% to 0.57 MPa compared to the one with 100% WF (A1, 1.02 MPa). The plywood prepared by UF resin A5 showed the best wet shear strength (1.34 MPa), which improved by 31.4% compared to that of Resin A1. This could be due to the micro-fibrous morphology of SEP, which could make the UF resin molecules gather around SEP during the curing process, thus leading UF resin molecule distribution to be more homogeneous and forming a higher dense structure. Another possible reason was that the SEP with fibrous morphology could form an interpenetrated network with the UF resin system, which further improved the wet shear strength of the produced plywood.

### 3.2. Formaldehyde Emission Measurement

In addition to the bonding strength, formaldehyde emission is another important property for the practical application of plywood bonded by UF resins. The formaldehyde emissions of plywood prepared by different UF adhesives are shown in Figure 2. The formaldehyde emission of plywood bonded by UF Resin A1 was 1.23 mg/L. When adding 20% SEP, the formaldehyde emission of plywood bonded by the developed Resin (A2) decreased by 36% to 0.79 mg/L. This was attributed to the free formaldehyde adsorption capacity of SEP. Because of the large specific surface area and special tunnel structure, the SEP could capture free formaldehyde into itself, which reduced the free formaldehyde content of the resin and formaldehyde emission of the resulting plywood. However, as the SEP addition further increased, the formaldehyde emission of plywood began to increase. For Resin A5, the formaldehyde emission increased by 22% to 1.50 mg/L compared to the plywood bonded by Resin A1 (1.23 mg/L). SEP adsorption characteristics are only a simple physical process, which indicates that the SEP adsorption capacity is limited. From the other perspective, this tunnel structure of SEP could accelerate the free formaldehyde release, which was a dominating factor for the plywood formaldehyde emission. As a result, formaldehyde emission was increased when the SEP addition exceeded 40%.

Based on the research of Ding et al. [28] and He et al. [29], the formaldehyde mainly emitted through vessels of the veneer. Therefore, the SEP tunnel structure of the inner layer is more conducive to the further release of formaldehyde of the resultant plywood over a long time. Furthermore, the formaldehyde emission of the resulting plywood specimens was retested after 30 days in this research. With the elongation of the storage time, the formaldehyde emission of the resin layer with tunnel structure should be faster than that without SEP.

The formaldehyde emission of the plywood after 30 days was evaluated and is shown in Table 2. The results showed that the declined rate of the samples increased as the SEP ratio increased in the filler, which indicated that the tunnel structure of SEP was beneficial to the releasing of formaldehyde. For UF Resin A5, the formaldehyde emission of the resulting plywood (1.50 mg/L) was 18% higher than that of UF Resin A1 (1.23 mg/L). After 30 days, the formaldehyde emission of the plywood bonded by UF Resin A1 and UF Resin A5 were 1.03 and 0.58 mg/L, reducing by 16.9% and 61.3%, respectively. Moreover, the final formaldehyde emission of plywood bonded by UF Resin A5 was 43.7% lower than that with Resin A1. In the plywood fabrication industry, the resulting plywood undergoes a hot stacking process for three to four weeks to cool down, which is beneficial to release formaldehyde, especially for the plywood bonded by UF resin with SEP. The schematic model of tunnel-structured SEP and the absorption and release method of formaldehyde are shown in Figure 3.

### 3.3. FTIR Analysis

Figure 4 shows the FTIR spectra of different resins. The peak at 3349.05 cm^−1^ was assigned to the stretching vibration of N–H and O–H bonds in the primary amines and hydroxyl groups [30]. The flexing vibration of the C–H of methylene occurring at 2962.69 [31], 1601.63, and 1550.80 cm^−1^ were acylamino absorption bands; the peaks at 1388.08 and 1238.47 cm^−1^ were the deformation vibrations of the plane of methylene; the peak near 1135 cm^−1^ was the absorption band of ether linkage of CH_2_OCH_2_ [32] and 1001.59 cm^−1^ was the absorption band of the methylol of CH_2_O. Specifically, when SEP was added, the ether bond of the UF Resin A5 shifted toward a low wavenumber (from 1135.70 to 1131.32 cm^−1^, red shift), which could be attributed to the hydrogen bonding interactions between the –OH on SEP and the ether bond of the UF resin [33].

### 3.4. TG and DTG Analysis

Figure 5 shows the TG and DTG curves of different resins. The thermal degradation process of UF resin could be divided into three stages. The first stage was at a temperature region of 20–120 °C, which was attributed to the evaporation of water [34]. The second stage started at 200 °C and ended at 260 °C, which could result from the loss of small molecules and the break of unstable chemical bonds, such as ether bond, methylene [35]. The mass loss at the second stage was 11.43% for Resin A5 and 19.00% for Resin A1, respectively. The third stage started at 260 °C and ended at 350 °C, indicating the resin structure degradation. The mass loss at the third stage was 55.56% for Resin A5 and 58.61% for Resin A1, respectively. The residue was 31.28% for Resin A5 and 19.82% for Resin A1, respectively.

The TG curves showed that the degradation trend of the Resin A5 was similar to that of Resin A1. By contrast, the small mass loss and the higher residue amount of Resin A5 was mainly attributed to the high temperature resistance of SEP [36]. DTG results (Figure 5b) showed that Resin A5 and A1 almost had the same maximum weight loss temperature at 283.99 °C. The TG and DTG results indicated that SEP only mildly affected the thermostability of the resultant resin.

### 3.5. SEM Analysis

The morphology of fillers (WF and SEP) and fracture surface micrographs of cured resins are presented in Figure 6. WF was composed of ellipsoid starch granules with a smooth surface. SEP was composed of irregular fiber-like aggregates. The fibrous morphology could increase the interface area of SEP with the polymer matrix, which would result in strong interaction and reinforced polymers. Because of the inherent brittleness of the UF resin, UF Resin A1 exhibited a neat brittle fracture characteristic (Figure 6c). On the contrary, the fracture surface of UF Resin A5 became rough and appeared wrinkled, indicating the toughness of the cured resin layer increased. The increased toughness may be due to the interpenetrated network formed by the fibrous SEP and the UF resin system. Thus, when subjected to external forces, the tough adhesive layer acted as a buffer, resulting in an uneven fracture surface with wrinkles. This also explained the reason why the wet shear strength of plywood prepared by UF Resin A5 was improved by 31.4% compared with that of UF resin without SEP.

### 3.6. Water Contact Angle Analysis

The contact angle test results in 5 s of the cured Resin A5 and A1 film are presented in Figure 7. The water contact angles of Resin A1 and Resin A5 were 45.44° and 42.18°. Addition of SEP into UF resin decreased the contact angle, which was due to the natural hydrophilic properties of SEP [37]. From the wet shear strength analysis (Section 3.1), the plywood prepared by Resin A5 has the optimal strength, indicating that the hydrophilic property of SEP did not negatively affect the other properties of the UF resin.

## 4. Conclusions

When UF resin with 80% SEP + 20% WF filler (Resin A5) was used, the wet shear strength of the resulting plywood improved by 31.4%, which was attributed to the interpenetrated network formed by the fibrous SEP and the UF resin system, which resulted in a toughened fracture of the cured adhesive layer and further enhanced the bonding strength of the plywood.

Consistent with assumptions, the tunnel structure of SEP is beneficial to release formaldehyde, which was verified by the formaldehyde emission test after 30 days, i.e., the formaldehyde emission of the plywood bonded by UF Resin A5 declined by 43.7% compared with that of Resin A1.

Considering the wet shear strength, the formaldehyde emission, and cost of the plywood, the best strategy to modify the UF resin was the addition of 80% SEP + 20% WF filler.

## Figures and Tables

**Figure 1 polymers-11-01286-f001:**
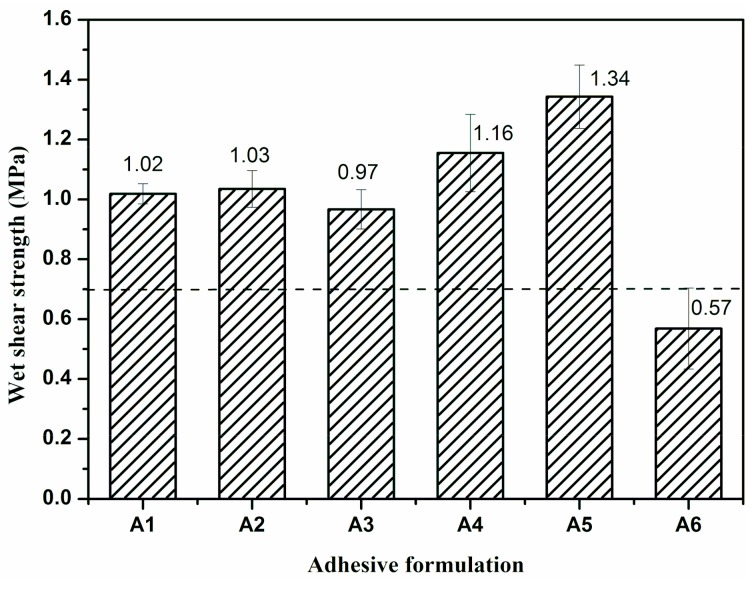
The wet shear strength of plywood bonded by urea formaldehyde (UF) resin with different fillers: Resin A1—0% SEP + 100% WF, Resin A2—20% SEP + 80% WF, Resin A3—40% SEP + 60% WF, Resin A4—60% SEP + 40% WF, Resin A5—80% SEP + 20% WF, Resin A6—100% SEP + 0%WF. Error bars represent standard deviations from eight replicates.

**Figure 2 polymers-11-01286-f002:**
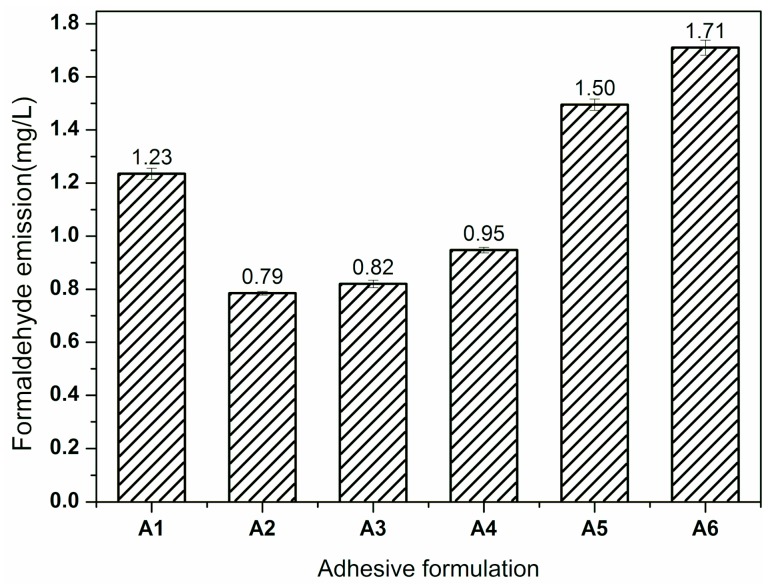
The formaldehyde emission of the plywood bonded by UF resin with different fillers: Resin A1—0% SEP + 100% WF, Resin A2—20% SEP + 80% WF, Resin A3—40% SEP + 60% WF, Resin A4—60% SEP + 40% WF, Resin A5—80% SEP + 20% WF, Resin A6—100% SEP + 0%WF. Error bars represent standard deviations from three replicates.

**Figure 3 polymers-11-01286-f003:**
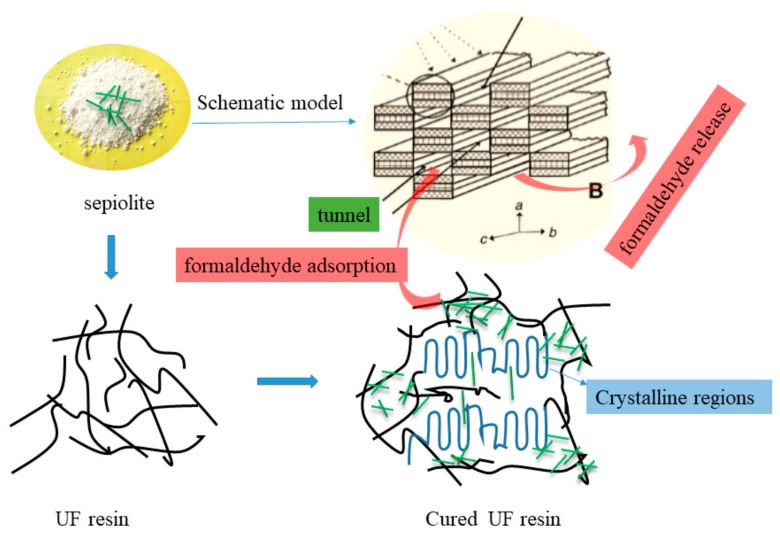
Schematic model of tunnel-structured SEP and the absorption and release method of formaldehyde.

**Figure 4 polymers-11-01286-f004:**
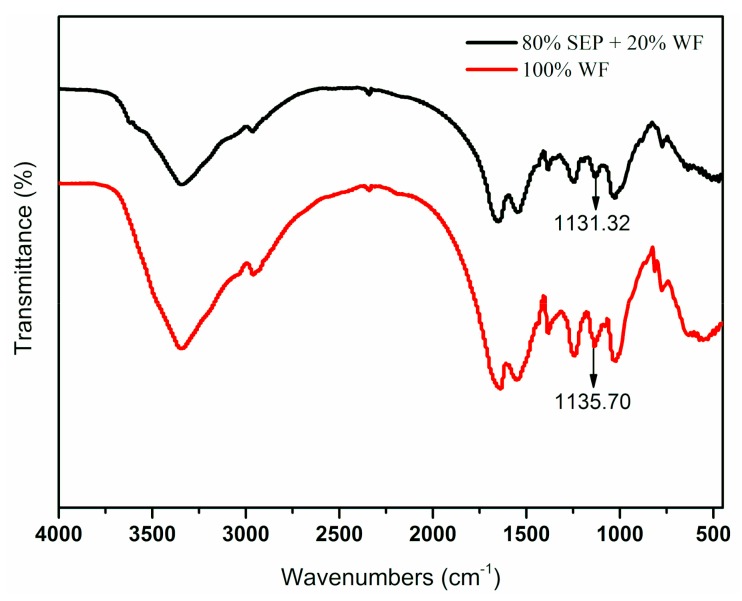
FTIR spectra of UF resin with different fillers: Resin A1—100% WF + 0% SEP; Resin A5—80% SEP + 20% WF.

**Figure 5 polymers-11-01286-f005:**
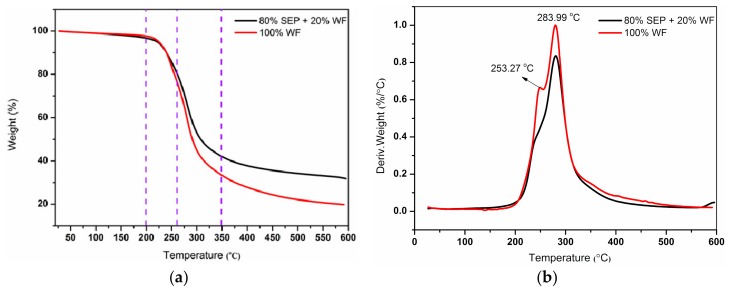
TG (**a**) and DTG (**b**) curves of UF resin with different fillers: Resin A1—100% WF + 0% SEP; Resin A5—80% SEP + 20% WF.

**Figure 6 polymers-11-01286-f006:**
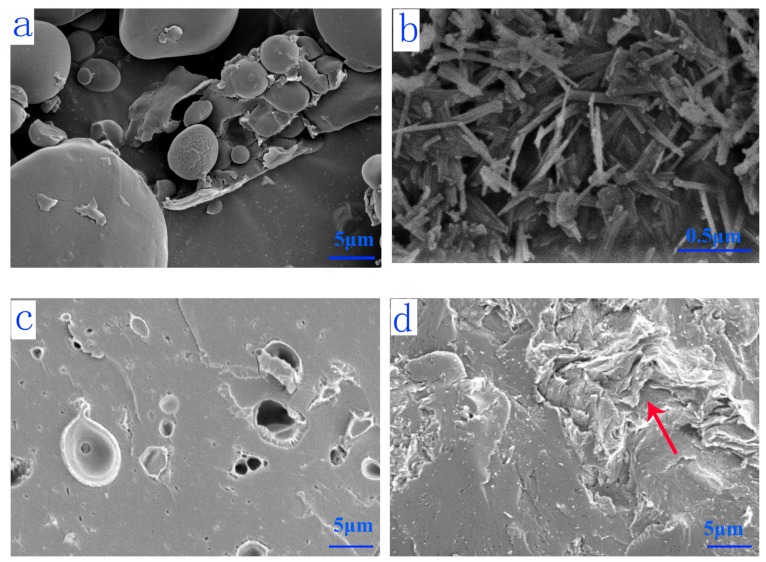
SEM images of WF (**a**), SEP (**b**), and cured UF resin with different fillers: Resin A1—100% WF + 0% SEP (**c**); Resin A5—80% SEP + 20% WF (**d**).

**Figure 7 polymers-11-01286-f007:**
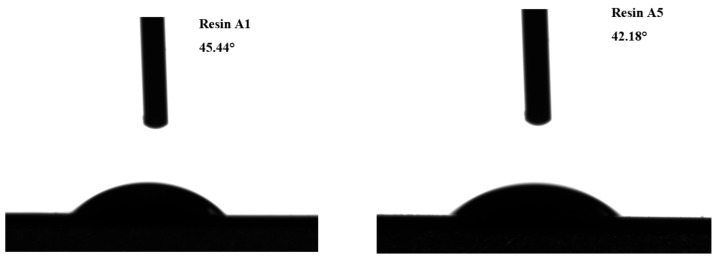
Water contact angles of Resin A1 and Resin A5.

**Table 1 polymers-11-01286-t001:** The different formulations of sepiolite (SEP)-based fillers.

Resin No.	SEP (%)	WF (g)	SEP (g)
A1 (0% SEP + 100% WF)	0	25	0
A2 (20% SEP + 80% WF)	20	20	5
A3 (40% SEP + 60% WF)	40	15	10
A4 (60% SEP + 40% WF)	60	10	15
A5 (80% SEP + 20% WF)	80	5	20
A6 (100% SEP + 0% WF)	100	0	25

WF—wheat flour.

**Table 2 polymers-11-01286-t002:** The comparison of the plywood initial formaldehyde emission and the values after 30 days.

Samples	Initial Formaldehyde Emission (mg/L)	After 30 Days (mg/L)	Decline Rate (%)
A1	1.24	1.03	16.9
A2	0.79	0.65	17.7
A3	0.82	0.63	23.2
A4	0.95	0.69	27.4
A5	1.50	0.58	61.3
A6	1.71	1.44	15.8

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
