# Peer review of "Urea Formaldehyde Resin Resultant Plywood with Rapid Formaldehyde Release Modified by Tunnel-Structured Sepiolite"

_polymers, 2019, doi:10.3390/polym11081286_

Round 1

Reviewer 1 Report

This manuscript provides relevant results on formaldehyde emission reduction when plywood is bound with sepiolite combined with wood flour. However, I do not believe that the amount and depth of scientific content provided merits publication in a high ranking journal like Polymers.

Some specific comments below.

Even though it is acceptable that SEP may act as a formaldehyde scavenger, due to its adsorption capacity, and as mechanical reinforcement, the idea that its tunnel structure “provides a channel for gas escape” is unfounded. The authors provide no evidence of support this. In addition, why does formulation A6, with the highest SEP content, have the lowest emission decline rate after 30 days?

Page 5 – “Except for…” should be “In addition to…”

Page 6 – There is no explanation for why more unstable ether bonds were formed in resin A5.

Page 7 – The hypothesis that the smaller mass loss in resin A5 is due to addition of SEP “reducing the formation of unstable chemical bonds during the resin curing process” is awkward. Why is this not simply due to SEP being thermally stable, as the authors point out when discussing the higher final residue?

The authors do not provide a recommendation for what would be the best SEP content. Formulation A2 seems to be the most interesting in terms of low formaldehyde emission, despite shear strength not being improved. Or should one take into account only the emission values after 30 days storage?

Reviewer 2 Report

Its an interesting article. Though the author characterized the materials by different techniques, however its also need to improve the manuscript before publication. 

All the analysis need to explain in details, not only mentioned the result.

As they synthesize a new product, they need to analyze the NMR, FT-IR is not enough to conclude a synthesis process, please also add the UF as well as urea and formaldehyde in both FT-IR and NMR, this will help the reader clear synthesis process. 

The author mentioned water resistance measurement, but its actually wet shear strength. Please correct it in analysis section. 

Its also need to check the water resistance by water swelling and water contact angle test.

Without TEM analysis the morphology characterization is incomplete, its need to run TEM analysis.   

Round 2

Reviewer 1 Report

I remain doubtful about the relevance of this manuscript for publication in Polymers, despite the improvements made in this revised version. The paper lacks somewhat in depth and extent of scientific content.

One particular comment. The discussion of TG results is still inappropriate. After addition of SEP the TG curve did not "move to the high-temperature direction" or delay decomposition as the authors say. The DTG graph shows that the two DTG peaks remain centered at the same temperatures. So SEP does not seem to improve thermal stability of the resin. What happens is that the final residue is higher, due to the presence of undegraded SEP!

Reviewer 2 Report

The author responded and revised properly. However I still suggests to considering water contact angle.  

Round 3

Reviewer 1 Report

I have no more specific comments.